# Transport, Associated Handling Procedures and Behaviour of Calves Marketed through Chilean Auction Markets

**DOI:** 10.3390/ani10112170

**Published:** 2020-11-21

**Authors:** Viviana M. Bravo, Toby G. Knowles, Carmen Gallo

**Affiliations:** 1Programa Doctorado en Ciencias Veterinarias, Escuela de Graduados, Facultad de Ciencias Veterinarias, Universidad Austral de Chile, Casilla 567, Valdivia 5090000, Chile; 2School of Veterinary Science, University of Bristol, Langford, Bristol BS40 5DU, UK; toby.knowles@bristol.ac.uk; 3Instituto de Ciencia Animal and OIE Collaborating Centre for Animal Welfare and Livestock Production Systems, Facultad de Ciencias Veterinarias, Universidad Austral de Chile, Casilla 567, Valdivia 5090000, Chile; cgallo@uach.cl

**Keywords:** transport, calves, marketing, behaviour, handling, welfare

## Abstract

**Simple Summary:**

Cattle are transported at least once in their lives, primarily associated with marketing. In Chile one of the main channels for selling animals is auctions. At auction markets, the procedures associated with transport of livestock (loading, unloading and travel) are at least duplicated, affecting their welfare. The aims of this study were to describe the procedures associated with transport and handling of calves (cattle < 9 months old) during marketing through auctions, and to evaluate compliance with Chilean law. A survey of drivers who transport calves from originating farms to markets (OM) and from markets to destination farms (MD) was performed. During loading and unloading, handling by stockpersons, facilities, calf fitness for transport, and behaviour were evaluated using protocols. Most drivers claimed having the training required by law, used bedding material, and provided adequate space for the animals; most OM and MD journeys were less than 2 h. Most calves were assessed as fit for transport. Calves slipped often and some practices associated with poor handling were still observed, mainly during loading. Compliance with the law during transport of calves was adhered to; however, associated handling during unloading and loading is still inadequate, evidencing a need for training.

**Abstract:**

In Chile, selling animals through livestock markets is common. At markets, stressful events like loading, unloading and travel are at least duplicated. We described procedures associated with transport of calves at 20 markets and evaluated compliance with Chilean law by performing a survey of drivers who transport calves from origin farms to markets (OM) and from markets to destination (MD). During loading and unloading, we evaluated handling by stockpersons, facilities, fitness for transport, and behavioural indicators of the calves through direct observation using protocols. A total of 80% of drivers claimed having the training required by law. The mean travel time was 1 h 31 min for OM and 1 h 44 min for MD journeys (overall range 5 min–40.5 h). Most drivers used bedding material and provided adequate space availability. A total of 99.2% of the observed calves were assessed as fit to transport; slipping, turning back, vocalizing and balking were frequent behaviours observed during loading and unloading. Prohibited practices like prodding and hitting using driving devices were still observed, mainly during loading. Compliance with the law during transport of calves was adhered to; however, the associated handling within markets was still inadequate, evidencing need for training in order to improve animal welfare.

## 1. Introduction

Cattle are transported at least once in their lives, and this transportation is usually linked with marketing. The marketing process is inherently stressful for animals because it involves handling, transportation and penning/housing in unfamiliar places [1]. In Chile, transportation is often associated with prolonged fasting times, since cattle are deprived of food and water from the moment they are herded before being transported until they arrive at destination; lorries are not equipped to provide food or water to cattle during long journeys (>24 h) and cattle are not commonly unloaded at resting posts [2]. 

Cattle producers can be divided in those who breed and fatten beef (complete cycle), those who only produce weaned calves (cattle < 250 kg) and those who buy calves for fattening [3]. In the two latter cases, calves must be sold/bought, and auction markets are one of the most popular marketing channels within Chile. During 2018, 940,397 cattle were auctioned in Chilean markets, 37% of which were calves [4]. When animals are marketed through auctions, transport and fasting times are extended, and the times that they are loaded, unloaded, handled, held in unfamiliar surroundings and mixed with unfamiliar animals is multiplied several times [5]. Handling and facility features have been reported to affect the welfare of calves within Chilean markets [6]. In the case of beef calves, weaning stress is added, because this process is often carried out just before loading for sale. Weaning followed by transport are considered as maximal stressors for beef calves [7]. 

Several factors separately or combined can determine the welfare of cattle during transportation, including driver experience and training, quality of handling, stocking density, duration of transport and rest stops, truck design, animal behaviour features, road and environmental conditions and fitness of the animals for travel [8]. The current Chilean regulation on the protection of animals during transport regulates features related with handling during loading and unloading and mandates the presence of a competent person to oversee the welfare of animals during the whole process; this person is usually the driver himself, who should have attended an official training course related to animal welfare during transport [9]. The aims of this study were to describe transport and associated handling procedures of calves marketed through Chilean auction markets, to evaluate compliance with Chilean Government regulations and to assess handling by stockpersons, facilities, fitness for transport and behavioural indicators of the calves during unloading and loading.

## 2. Materials and Methods

The Bioethics Committee “Use of animals in research” of the Universidad Austral de Chile approved the present study (Application N° 325/2018). It was carried out between November 2017 (spring) and April 2018 (autumn). To evaluate the features of transport of calves that are marketed in Chilean auctions, twenty-one premises corresponding to all the markets located in the southern regions of the country (Geographical coordinates: −37.81208° S–72.67112° W to −45.61736° S–72.10496° W), where most cattle are produced, were visited. We focused our study on weaned calves between 150 and 250 kg. A minimum of two observers trained in animal welfare and behaviour were present at each visit.

### 2.1. Driver Training and Journey Conditions

A survey of the truck drivers in charge of transporting calves from farms of origin to markets (OM) and from markets to destination farms (MD) was carried out. The survey was completed by one of the observers during the arrival/departure of trucks at the unloading/loading area. Questions sought to discover: level of driver training according to the Chilean law (present-absent), planning of journey beforehand (present-absent), possession of a contingency plan to face possible emergency situations during the journey (present-absent), category and number of animals transported, travel time and type of road travelled, type of bedding used and truck space availability for cattle transport (m^2^). The OM travel time was calculated to include time of loading at the farm of origin, as indicated by the driver, and the time of unloading at markets. The MD travel time was also provided by the driver based on the loading time at the market and the estimated travel time until unloading at destination. The training of the truck driver was evaluated according to the requirements of current Chilean law [9]. Space availability was calculated in m^2^ per animal, according to the space availability of the truck (measuring length × width of truck loading area) and the number of animals transported.

### 2.2. Welfare Indicators during Unloading and Loading Procedures

As part of the transport process, the courses of unloading and loading in markets were assessed. The number of individuals unloaded/loaded per transport load was recorded. For each load we observed behavioural indicators of the calves, good/negative practices of truck drivers and market staff, and facility features. We used a protocol based on the scientific literature [10,11,12,13], current Chilean law [9] and previous test visits to markets, where we were able to identify the time and order of events associated with transport during marketing through Chilean auctions, including length and location of every procedure carried out. To avoid disagreement between observers, all observations regarding unloading and loading procedures were made by the same observer for all visits.

#### 2.2.1. Behavioural Indicators of the Calves

Behavioural events that have been previously linked with handling and/or facilities features under commercial conditions in cattle [6,10,13] were observed and recorded. We quantified slips, falls, jumps, balks, turns, vocalizations, mounts, aggressions and defecation/urination in each observed load of animals. Definitions of behavioural indicators used [11,14,15,16], are presented in Table 1.

#### 2.2.2. Good/Negative Practices of Truck Drivers and Stockpersons

It was recorded whether the truck driver or market staff were in charge of the procedure of unloading and loading the calves and the type of device used to move the calves. The ‘good practice’ features evaluated during unloading and loading were the presence or absence of: opening of both doors of the truck before unloading/loading animals; appropriate truck parking against the ramp avoiding gaps and sideward deviations; appropriate truck levelling with unloading/loading facilities (<20 cm difference between the truck height and the ramp height); the presence of obstacles or distractions that could interfere/stop the movement of the animals through pathways; whether calves were mixed with other species; and appropriate handling. Handling was described as appropriate if it was conducted quietly and without unnecessary noise, harassment or force. The occurrence of prohibited practices like striking (when the handler hits an animal using his driving device), pokes (when the handler pushes an animal using his driving device), kicks (handler beats the animal with a foot) and tail bending or pulling (handler twists or pulls the tail of the animal) was counted and recorded per load of calves.

#### 2.2.3. Market Facilities Features

Ramps used to unload/load the animals in markets were evaluated. Ramp slopes were obtained dividing the height by the length of the ramp then multiplied by 100; measurements were made using a tape. Ramp slope was considered as appropriate when it was ≤14 degrees (25%), according to the current Chilean legislation [9]. Floor type of the ramp (slippery/slip-proof), the presence/absence of side protections and of sharp surfaces that might harm the animals and appropriate lighting were recorded. Lighting was described as inappropriate when shadows or darkened places/raceways interrupted the movement of the calves.

### 2.3. Fitness/Health of Calves Marketed

During unloading we evaluated fitness of calves arriving at the market by counting and recording the number of lame, downer and/or dead animals per load. To evaluate the general health status of the calves during marketing, three to four pens with calves were selected for each market; through scan sampling, we assessed the presence/absence of the following indicators per pen: nasal and eye discharge, diarrhoea signs and skin lesions. All these observations were carried out by the same veterinarian on each visit.

### 2.4. Data Analysis

Qualitative data are presented as frequencies and quantitative variables using means (±SEM) and ranges. IBM SPSS Statistics Software Version 25 [17] was used to perform the analysis for the descriptive statistics. Comparisons of behavioural and handling variables between unloading and loading procedures were performed using chi square tests with Yate’s continuity correction and Fisher’s exact test, using the R program [18].

## 3. Results

One of the twenty-one markets visited did not receive any animals the day of our visit, so we were unable to perform surveys or observations. Shortly after, this market ceased its activities permanently due to the scarcity of animals. Hence, all results shown correspond to 20 markets visited.

### 3.1. Driver Training and Journey Conditions

A total of 190 OM surveys, corresponding to the transport of 1940 calves, and 118 MD surveys of 1688 calves were obtained. Surveys were carried out during unloading between 7:44 and 15:00 h; and during loading between 14:33 and 20:07 h. In one market, surveys were taken between 9:29 and 12:00 h the next morning because animals remained in the pens overnight and were loaded the day after the auction; we could not perform MD surveys in one of the markets because the loading process started late at night.

All results shown were based on the answers of the truck drivers, but in most cases it was also possible to visually corroborate number and animal category, and also truck features, including bedding type, by the observer who conducted the survey.

Some drivers carried calves OM and MD, and also across different markets, so they could have been surveyed more than once. Considering OM and MD, the survey was applied to 286 different drivers and 80% of them attested having the training required by Chilean law [9]. In addition, 74.1% stated that they planned their journeys beforehand; and 56.6% claimed to have a contingency plan to face possible emergency situations.

For OM journeys, the mean travel time was 1 h 31 min, (SEM ± 0.09) with a range of 5 min to 13 h. In the case of MD journeys, an average time of 1 h 44 min (SEM ± 0.49) with a range of 5 min to 40.5 h was obtained (Figure 1). The mode of travel time in both OM and MD journeys was 1 h.

Drivers declared traveling on paved roads only in 51.4% (94/183) of the OM cases and in 72.8% (83/114) of the MD cases; on mixed roads, including a paved and a non-paved section (gravel/soil) in 42.6% (78/183) of the OM and in 24.6% (28/114) of the MD cases; and just on non-paved roads in 4.9% (9/183) of the OM and 0.9% (1/114) of the MD cases. Additionally, in four cases (two OM and two MD) the drivers reported sea transport as part of the journey, which included the longest travel time reported (40.5 h).

The most frequently used bedding material was sawdust (59% OM; 57% MD), followed by sand (12% OM; 7% MD) (Table 2). A higher proportion of trucks without bedding was found in MD journeys (31.3%) compared with OM (15.4%) journeys (*p* = 0.0019). In 75% of the OM and 40% of the MD journeys without bedding, a rubber mat was used instead.

The mean space availability during OM journeys was 1.63 m^2^ per animal (SEM ± 0.15) and during MD journeys was 1.77 m^2^ per animal (SEM ± 0.17).

### 3.2. Welfare Indicators during Unloading and Loading Procedures

A total of 177 unloading and 111 loading procedures, corresponding to 1822 calves and 1493 calves, respectively, were assessed. Unloading was observed between 7:48 and 13:56 h (98.9% of the cases) and loading between 14:30 and 20:09 h (93.7% of the cases). Loading of calves during the morning was observed in one market, because most animals were loaded and transported the day after auction.

We found that in 6% of the unloading and loadings (17/288), calves were transported mixed with other species, including horses (n = 12), sheep (n = 5) and pigs (n = 1). We were not able to perform loading observations at one market for logistical reasons. The arrival of one group of calves on foot (walking) and the loading of one calf from one truck to another were also observed. Both cases were left out of the study.

#### 3.2.1. Behavioural Indicators of the Calves

Minimum, maximum and average percentage of behavioural indicators of the calves observed during unloading and loading are presented in Table 3. We observed an average of 18 calves per load (from 1 to 40 calves) during unloading and 24 calves per load (from 1 to 51 calves) during loading.

#### 3.2.2. Good/Negative Practices of Truck Drivers and Handling by Stockpersons

Unloading and loading was carried out only by drivers, only by market staff, or simultaneously by both. Truck drivers were involved in a significantly higher proportion of unloading (56%) than loading (36%) (*p* < 0.01), whereas market staff intervened in a significantly higher proportion of loading (74%) compared with unloading (49%) (*p* < 0.001).

Most unloading (59.7%) occurred with the use of no device at all, just by opening the truck doors and releasing the animals. Wooden sticks were the most common devices used to drive calves during unloading and loading (Table 4). Within “others”, we observed one wooden stick with a plastic bottle fitted at the end, ropes and the poles used as separators for cattle within the truck.

The results related to good and prohibited practices carried out by truck drivers and/or market staff during unloading and loading in markets are shown in Table 5.

The absence of obstacles was considered within the good practices of drivers instead of a facility-related variable, due to the fact that in 91.5% and 89.6% of the cases, the objects identified obstructing the path of the calves during unloading and loading, respectively, were the removed poles used as separators within the truck, or/and the position of handlers themselves in front of the animals to be unloaded. In fact, in 29% of the unloading observed, it was found that the truck separators (poles) were thrown by the truck driver inside the truck, underneath the animals, before they exited the vehicle; hence acting as obstacles (Figure 2).

In terms of the handling of the animals by stockpersons, the presence of hits and pokes with the devices used for driving the calves was frequently observed (Table 5) and was significantly higher during loading than during unloading. Regarding prohibited practices (percentage of negative tactile interactions per group of calves), we recorded (average percentage) that during unloading and loading, respectively, 18% and 92% of the calves were hit (beaten), 30% and 220% were poked with the driving devices, 3% and 7% were kicked, and 0% and 1% were tail bent.

#### 3.2.3. Market Facility Features

Facilities used for loading and unloading animals were the same for both procedures: 73% of the procedures were performed using ramps with slopes of ≤14°, 99% with non-slippery floor, 98% with appropriate lighting, and 100% with side protections and an absence of sharp surfaces.

### 3.3. Fitness/Health of Calves Marketed

During unloading, we were able to evaluate 1822 calves for fitness and recorded the arrival of 10 downers (0.55%), 2 dead (0.11%) and 2 lame calves (0.11%). One lame calf was observed during loading procedures (0.07%). Of a total of 1477 calves visually evaluated for the presence of health problems in a total of 66 different pens at the 20 markets, we found 54 calves with eye discharge (3.6%); 3 with nasal discharge (0.2%); 12 calves with skin lesions (0.8%) and no calves with diarrhoea signs.

## 4. Discussion

### 4.1. Driver Training and Journey Features

The current Chilean legislation related to the protection of animals during transport establishes that there must be a person responsible for the welfare of livestock during the journey, loading and unloading procedures [9]. This person must have knowledge related to the behaviour and needs of the animals and should be able to handle them efficiently, preserving their welfare. Therefore, truck drivers must comply with these requirements and should have a certificate of approval of an official training course, recognized by the competent authority (Agriculture and Livestock Service), unless they are technicians or professionals linked to the livestock industry. According to our results, 80% of the surveyed truck drivers attested to having the required training; this is a high proportion, considering that the Chilean law [9] was passed in 2013, but has only been subject to enforcement since 2016. The Chilean legislation [9], as well as the OIE Terrestrial Animal Health Code [19], also indicates that drivers are responsible for planning the journey to ensure the care of the animals, including developing and keeping up-to-date contingency plans to address emergencies, and these issues must be included in the training courses. The fact that we found that 74.1% of the drivers surveyed stated that they planned their journeys, and 56.6% mentioned having a contingency plan to face emergency situations indicates that the training courses actually include these contents, and that the drivers are acting accordingly, which should lead towards improving animal welfare during transport.

Although our results showed a short average time of travel in both OM and MD journeys, the variation was high (5 min to 40.5 h). Marketing cattle through auction markets at least doubles the number of times that they are loaded, transported and unloaded, and also prolongs times of food and water deprivation compared with selling the calves directly from one producer to another. Cattle sold through markets were described as being thirstier and more tired on arrival at the lairage in the abattoir than cattle sent direct from farms [20]. According to our observations, calves remain approximately 12 h at markets, arriving early in the morning and departing at the end of the day; they do not have access to food or water during their permanence at the markets, unless they have to remain more than 24 h in the premises. In a preliminary study in Chile, we evaluated the effect of 3 h transportation followed by a 21 h fasting period in weaned calves of similar age and weight, finding that calves lost 6.8% of live weight in the 24 h of food and water deprivation [21]. Moreover, the higher concentration of blood betahydroxibutyrate found in the same calves after fasting compared with before loading, suggests that within this time frame, there was not only a loss of gut content in the calves, but fat was also mobilized from body tissues. These live weight losses are important for the welfare of the calves marketed, because they probably suffer from hunger and thirst during this process. The economics of the sales may be affected because animals are sold per kg live weight. This becomes particularly important in the cases of calves that are transported for over 24 h after marketing to their destination farms, as in the 40.5 h MD journeys recorded. The recovery period for weight and blood stress indicators in calves after such long-distance transportations has been shown to be close to a month [22].

In South American countries, there are paved carriageways in good condition leading to the main cities, but there are also many unpaved or stone roads, often in bad condition; this is especially the case for the side roads serving the farms [3]. As most drivers surveyed declared travelling only on paved roads, which are smoother, road type should not represent a major welfare problem for most of the calves marketed. A study carried out with sheep showed that transport on unpaved, rough roads had a greater detrimental effect on stress physiology compared with journeys on smoother roads [23]. However, four drivers also reported sea transport on ferries as part of the journey, which included the longest travel time reported (40.5 h). This type of transportation corresponds to a journey from the southernmost market in Chilean Patagonia to destination farms in the central, southern part of the country, involving both road and roll-on roll-off sea transport. The Patagonian regions of Chile are characterized by lack of good pasture for fattening, and hence many calves from this area are transported after weaning to fattening farms located further north, using this combination of maritime ferry and terrestrial routes [24]. It has been reported that this type of long-distance transport adversely affects the welfare of calves in terms of physiological and behavioural indicators, due to exposure to many more stressors related to ferry movement, inclement weather and inadequate water and feed provision (fasting or disruption of the normal pattern of feeding) [25].

The current Chilean legislation does not regulate the use of bedding during cattle transport, just indicating a requirement of non-slippery floor in trucks [9]. Even so, in 76% of all journeys, bedding was used, the material being mainly sawdust. The use of bedding material is recommended for comfort and insulation under cold conditions, particularly when transporting young stock. During the survey we noticed that, in addition, bedding was perceived by the drivers as being necessary to avoid slips and falls of calves during travel. González et al. [26], in a survey applied to drivers of long-haul transport (≥400 km) of cattle departing from, and arriving to the province of Alberta, Canada, found a significant association between the type of cattle being hauled and the use of bedding. These authors suggested a role of the economic value of cattle transported (e.g., breeding cattle) and of the perceived requirement or fragility (e.g., calves) in the use of bedding, which is in agreement with our findings. The proportion of trucks not using bedding was higher in MD cases; the reason for this may be that loading was carried out at the end of the auction, when stockpersons, including drivers, were probably tired and not willing to add bedding; they had already passed through the process of placing and changing bedding after arrival at the market (same drivers in OM and MD journeys). There is also the additional monetary cost of the bedding itself, as well as work required to add it, remove the soiled bedding, disposal, and cleaning the truck [8]. In 75% of the OM and 40% of the MD journeys without bedding, a rubber mat was used instead. According to the drivers, the rubber mats supply a soft, non-slippery floor for the transported animals and are easy to clean, saving time and money.

There is a strong economic motivation to load cattle as densely as possible due to the costs of transportation [8]. The Chilean legislation allows a maximum stocking density of 500 kg/m^2^ for cattle transport in general, not discriminating by category [27]. We calculated a mean space availability in OM and MD journeys of 1.63 m^2^ and 1.77 m^2^ per calf, respectively. Hence, even if we assume the maximum weight of 250 kg, the mean space availability was generally higher than the minimum required by law (1 m^2^ per 500 kg live weight). At high stocking densities, cattle occasionally go down, apparently involuntarily [8]. In this respect, the calculated space availability for the one journey where three downer calves arrived was only a third (0.56 m2 per animal) of the mean calculated for OM travels. However, the space availability calculated for one of the trucks arriving with one dead calf was 5.7 m^2^. Too little or too much space during cattle transportation may both lead to compromised animal welfare, particularly when other factors are also involved: in the latter case, the travel time was 13 h, and we do not know if the calf was observed to be fit for transport before loading. It is known that in longer journeys, animals get tired, increasing the number of lying animals and are also more prone to fall [28].

### 4.2. Welfare Indicators during Unloading and Loading Procedures

Mixing different species during transport represents a risk for the smaller, less aggressive animal species/categories. The Chilean regulation for cattle transport [27] dictates that animals with incompatible characteristics should be physically separated within the truck in these cases. We found that 6% of all observed unloading and loadings did not meet this requirement, because calves were even mixed with horses, sheep or pigs. According to our observations, this happened mainly with small trucks carrying only a few animals of each species, usually from small farmers. As there is no mention in any of the Chilean regulations on transport about specifically mixing animals of different species, separation of different species should be included in future changes of regulations.

Regarding facilities in general, current Chilean legislation regulates some features of ramps used during unloading and loading procedures [9]. We found that most loading and unloading was performed using ramps that fulfilled the requirements of having slopes ≤ 14°, slip-proof floors, side protection and an absence of sharp surfaces that might harm the animals. The lack of appropriate floor types has been described as a predictor variable for an increase in behaviours related to poor movement within auction markets [6]. Although generally calves were handled on non-slippery floors, some negative practices that can increase slips during unloading and turning back during loading could be due to incorrect use of the facilities, e.g., parking leaving gaps between the ramp and the truck, sidewards deviations, or even incorrect levelling by choosing the wrong ramp when different ramp heights were available at the markets, as frequently observed. There was still a level of carelessness by drivers who opened only one of the two back doors of the truck when unloading/loading the animals, and others who threw the security poles inside the truck before unloading, in order to have one less job to do afterwards.

During the recorded observations, market staff intervened in a higher proportion of loadings compared with unloading, where the use of devices to drive the animals was also higher compared with unloading. Wooden sticks and plastic pipes were the most frequently used devices to drive the calves and the sharpening of tips to poke animals was observed on both devices; in two cases, we witnessed nails inside the wooden sticks. In 6% and 1% of cases, flags were used to unload and load calves. Incorrect use of flags was associated with poking and hitting calves with the device. The use of a mixture of devices (more than one person involved in the procedure) and the use of no device was also recorded. The fact that loading is known to be more stressful for the animals than unloading, together with a lack of training of the stockpersons in charge of this operation are likely be the reasons for the increased use of devices and the handling problems observed during loading [6,29]. Although Chilean regulations state that there must be at least one trained animal handler in charge of livestock handling at each market, a shortage of trained and qualified staff among the stockpersons moving animals in the market has been previously described [29]. Managers often appear to be reluctant to invest too much effort in training stockpersons because of the high rate of turnover. The lack of continuity due to the fact that much of the staff working in markets do it as a part time job is an obstacle for workers receiving appropriate training and weakens any commitment and pride that people might feel toward their position [6]. De Vries mentioned that training focusing on the working groups of stockpersons is likely to improve human-animal relationships at livestock markets [29]. Training stockpersons in good cattle handling practices leads to better attitudes and behaviour toward animals [30].

Loading is known to be a more stressful procedure, with more adverse effects on cattle welfare, than unloading [11]. In agreement, we found significantly higher values for the indicators of poor handling, such as inappropriate driving, use of hits, pokes and pulls or flexed tails, during loading compared with unloading. Hitting and poking were identified as the main handling problems during this stage; moreover, these techniques are prohibited by the Chilean regulations for livestock transport and welfare [9]. Hitting is a painful stimulus that obviously has negative welfare consequences for the animals. For cattle (and pigs), aversive actions by humans including hits, slaps, and kicks by the stockpersons increase fear [31]. The behavioural responses of fear of humans by animals include escape–avoidance behaviour or aggressive behaviour to humans [32]. In a recent study at some of these markets, we found an association between the proportion of negative tactile interactions by handlers and noisy and inappropriate driving with the presentation of behaviours related to movement of calves through auction markets, like slips, falls, balks, turn, and jumps [6]. According to Grandin [33], slips and vocalizations of animals are acceptable up to 3%; an increase in these proportions would reflect poor welfare. During unloading of the calves, we found that slips (25.1%), followed by vocalizations (14.7%), were the most frequently observed, whereas slips (14.7%) and reversals (11.5%) were the most frequently observed during loading. Observations performed in cull cows during unloading and loading in the same auction markets showed similar results [34]. However, in both cases, the percentages of these behaviours were much higher than acceptable according to Grandin [33]. According to our findings, we consider this to be a welfare problem related to the behaviour of stockpersons rather than to facilities features. Therefore, training stockpersons to avoid negative tactile interactions and to use animal behaviour to their advantage, presents an opportunity to improve the welfare conditions of animals marketed through this channel. Taking into consideration that fear reactions of animals can affect both human and animal safety, and reduce worker comfort and time efficiency [35], training could also improve the welfare of workers.

### 4.3. Fitness/Health of Calves Marketed

One of the major problems related to the welfare of cattle during transport is associated with fitness of animals for travel; checking the fitness of the cattle before loading each truck and their transport history may help to reduce downer animals or mortality [8]. Calves are more likely to become non-ambulatory and/or die during transport compared with fat and feeder cattle [36]. The Chilean legislation [9] is based upon OIE standards [19] with respect to establishing which animals are not fit to travel. The finding of 10 downers, two lame and two dead calves at arrival would point towards problems related to the conditions during transport (travel time, stocking density, driving skills, etc.) rather than due to the assessment of fitness for travel on origin farms. However, we were not able to confirm whether the few problems found originated during transport or the calves were loaded with pre-existing conditions. Considering that 99.2% of the calves observed during unloading arrived fit at the market and that 95.4% of the calves observed in pens had an adequate body condition and were visually healthy, we can conclude that the health of the calves marketed did not impose a welfare problem.

## 5. Conclusions

Transport of calves associated with marketing through auctions follow most of the requirements established by the current Chilean legislation on the protection of animals. Most truck drivers stated having the required training, planned their journeys beforehand; had a contingency plan to face possible emergency situations. Most journeys were of short durations, bedding material was used, and adequate space availability was provided. Additionally, most of the observed calves were observed as fit/healthy to travel and to be marketed. Facilities used for unloading and loading were mostly in compliance with the law; however, there was some carelessness by truck drivers who used them inappropriately. We were able to observe a high proportion of behaviours like slipping, balking and vocalizations during unloading, as well as reversals during loading, which may reflect fear in the calves and indicates problems during these procedures. Moreover, prohibited practices like hitting and poking animals with the devices used for driving the calves were still being carried out in a high proportion of cases, mainly during loading. As this procedure is performed mostly by market staff, it exhibits a lack of training of stockpersons present in these establishments, which has consequences for the welfare of the calves.

## Figures and Tables

**Figure 1 animals-10-02170-f001:**
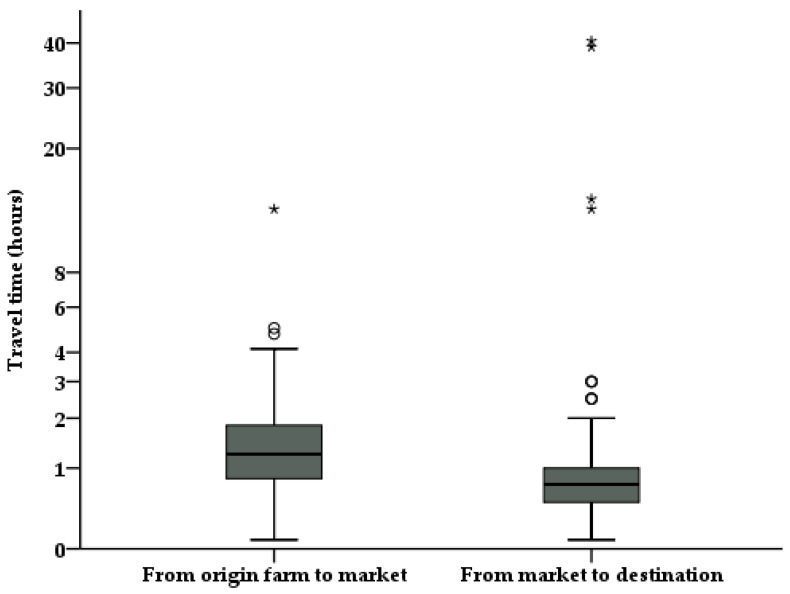
Travel time for calves marketed in livestock markets in the south of Chile according to drivers surveyed transporting calves from the farm of origin to the market and from market to destination farm. Circles and asterisks represent mild and extreme outliers, respectively.

**Figure 2 animals-10-02170-f002:**
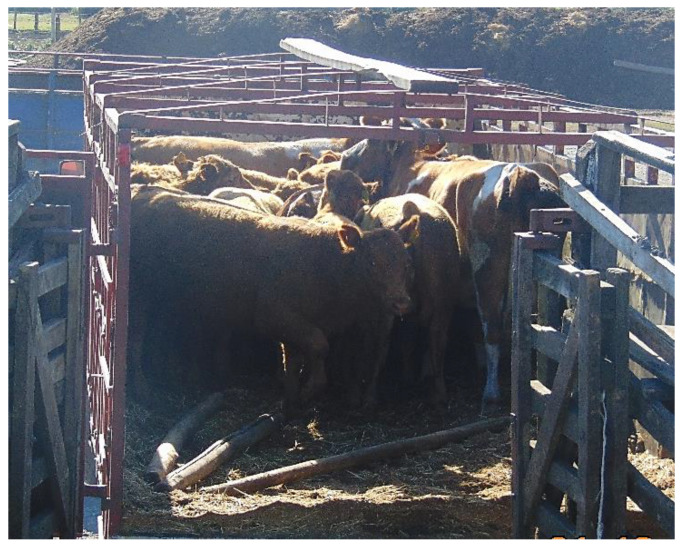
Separators (poles) inside the truck acting as an obstacle during unloading.

**Table 1 animals-10-02170-t001:** Definition of recorded behaviours in calves during unloading and loading.

Behavioural Indicators	Definition
Slip	A portion of the leg other than the foot, touches the ground or floor, or a foot loses contact with the ground or floor in a non-walking manner.
Fall	Loss of an upright position suddenly in which a part of the body other than the limbs touches the ground.
Jump	Leap with all four feet simultaneously off the ground in a manner or situation that could be hazardous.
Balk	A sudden stop or unwanted arrest of the animal flow.
Turn back	The calf changes direction of movement against the animal flow.
Vocalization	The calf makes a sound, emitted from the mouth (moos).
Mount	One calf stands on rear legs resting front legs and body on the back of another calf.
Aggression	Forcefully contacting with another calf using the head or any other part of the body, like hitting or pushing.
Defecation/urination	Elimination of faeces or urine from the body.

**Table 2 animals-10-02170-t002:** Bedding material used during transportation of calves from farms of origin to markets and from markets to destination farms.

Bedding	Origin Farms to Markets			Markets to Destination Farms		
Frequencies	Absolute Frequency	Valid Percentage	Cumulative Percentage	Absolute Frequency	Valid Percentage	Cumulative Percentage
Sawdust	108	59.3	59.3	65	56.5	56.5
Sand	21	11.5	70.8	8	7.0	63.5
Straw	14	7.7	78.5	2	1.7	65.2
Wood chips	7	3.8	82.3	2	1.7	66.9
Soil	2	1.1	83.4	2	1.7	68.6
Others	2	1	84.4	-	-	-
None	28	15.4	100	36	31.3	100
No data	8	-	-	3	-	-
Total	190	100	-	118	100	-

**Table 3 animals-10-02170-t003:** Minimum (Min). maximum (Max) and average (Aver) occurrence (%) of behavioural indicators of calves during unloading (177 loads) and during loading (111 loads) procedures.

	Unloading			Loading		
Behavioural Indicators	Min%	Max%	Aver%	Min%	Max%	Aver%
Slips	0.0	600.0	25.1	0.0	700.0	14.7
Falls	0.0	200.0	4.6	0.0	200.0	2.9
Jumps	0.0	100.0	5.4	0.0	100.0	7.7
Balks	0.0	333.3	10.2	0.0	100.0	6.7
Turns	0.0	33.3	0.6	0.0	500.0	11.5
Vocalization	0.0	700.0	14.7	0.0	200.0	3.3
Mounts	0.0	40.0	2.5	0.0	33.3	1.2
Agressions	0.0	100.0	0.8	0.0	3.4	0.0
Defecation/urination	0.0	33.3	0.9	0.0	300.0	5.2

**Table 4 animals-10-02170-t004:** Devices used to drive calves during unloading and loading in auction markets.

	Unloading			Loading		
Device	Absolute Frequency	Valid Percentage	Cumulative Percentage	Absolute Frequency	Valid Percentage	Cumulative Percentage
Wooden stick	44	27.7	27.7	60	55.6	55.6
Plastic pipe	6	3.8	31.5	21	19.4	75.0
Flag	4	2.5	34.0	1	0.9	75.9
Electric prod	1	0.6	34.6	3	2.8	78.7
Other devices	2	1.3	35.9	2	1.9	80.6
Mixed devices	7	4.4	40.3	12	11.1	91.7
None	95	59.7	100	9	8.3	100
No data	18	-	-	3	-	-
Total	177	100	-	111	100	-

**Table 5 animals-10-02170-t005:** Number of loads of calves observed (n), absolute frequency (AF) and percentage (%) of presence of good and prohibited practices carried out during unloading and loading in auction markets and confidence interval (CI) and *p*-value obtained by comparison between unloading and loading.

	Unloading			Loading					
Good Practices	N	AF	%	n	AF	%	CI		*p*-Value
Both truck doors open	174	92	52.9	109	59	54.1	−0.13939	0.11429	0.9335
Appropriate parking	175	118	67.4	108	76	70.4	−0.14753	0.08869	0.6995
Appropriate levelling	177	83	46.9	110	61	55.5	−0.21144	0.04021	0.1974
Absence of obstacles	172	77	44.8	109	68	62.4	−0.30111	−0.05124	<0.01 *
Appropriate driving	165	85	51.5	107	13	12.1	0.28773	0.49958	<0.001 *
**Prohibited Practices**									
Presence of hits	177	24	13.6	111	57	51.4	−0.49103	−0.26481	<0.001 *
Presence of pokes	177	42	23.7	111	74	66.7	−0.54450	−0.31426	<0.001 *
Presence of kicks	177	12	6.8	111	10	9.0	−0.09450	0.04991	0.6417
Presence pull/flex tail	177	1	0.6	111	6	5.4	0.00215	0.84280	0.0144 *

* Statistical differences (*p* < 0.05) between unloading and loading.

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
