# Peer review of "Transport, Associated Handling Procedures and Behaviour of Calves Marketed through Chilean Auction Markets"

_animals, 2020, doi:10.3390/ani10112170_

Round 1

Reviewer 1 Report

Manuscript:

Transport conditions of calves marketed through Chilean auction markets.

General comments

In this manuscript, the authors aimed to characterize the welfare of calves traded in Chilean auction markets, by describing the transport conditions (based in a survey carried out with the truck drivers) and the associated handling procedures (carried out during unloading and loading), checking if they are in compliance with Chilean Government regulations. The study (which can be characterized as descriptive) address a subject of practical interest and, despite of being carried out under a specific context (the Chilean condition), offers some methodological and results interpretation elements that are useful for who decide to carried out similar studies when assessing the welfare of cattle kept under similar conditions in other countries.

The objectives are well defined and, in general, the methodology used in the study is adequate to achieve them. However, some aspects of the methodology and results (as described below) need to be corrected or better explained. Special attention should be paid to avoid circular definitions.

The section that deserves the most attention from the authors is the Discussion, which needs to be completely revised. Much of the text presented in the discussion would be better characterized as a review, since they are not related to any of the results presented. It is also noteworthy that, in some cases, results from previous studies are presented that address characteristics that were not evaluated in the present study. Additionally, it is common to find paragraphs in the discussion that repeat the presentation of results, without the need to do so. Thus, I emphasize the need of rewritten the Discussion

Additional comments and suggestions are presented below.

Specific comments

Abstract

The abstract writing induces the reader to understanding that all data were obtained by surveying the truck drivers. Please, include information about the methodology used to assess the welfare indicators during unloading and loading and the behaviour of calves.

Introduction

L61. The content of the first phrase (“…The transport process…”) was already stated in the previous paragraph. I suggest removing it.

L69. According my understanding the study have two objectives. Thus, replace “The aim of this study was…” by “The aims of this study were…”

Material and methods

L95. Replace “…the course of unloading and loading in markets was assessed…” by “…the courses of unloading and loading in markets were assessed…”.

L99. Could you, please, be clearer about what you did when carrying out the “previous test visits to markets?

Table 1. Most of the definition of the behavioural categories assessed need to be improved, as follow:

Slip - Circular definition, consider using the NAMI definition (“...Slips occur when a portion of the leg other than the foot touches the ground or floor, or a foot loses contact with the ground or floor in a non-walking manner....", see: https://animalhandling.org/sites/default/files/forms/Animal_Handling_Guide091719.pd)

Fall - Again, I suggest using the NAMI definition ("...A fall occurs when an animal loses an upright position suddenly in which a part of the body other than the limbs touches the ground...", see: https://animalhandling.org/sites/default/files/forms/Animal_Handling_Guide091719.pdf)

Balk - How do you know that the animals balked due to "apparent distraction or intimidation"? Review.

Vocalization - Suggestion: "The calf makes a sound, emitted from the mouth (mooes)."

Mount - Circular definition, consider using the following definition: "...One calf stands on rear legs resting front legs and body on the back of another calf..." (adapted from Landford et al., 2011. https://doi.org/10.3168/jds.2010-3309).

Agression – Consider using: "Forcefully contacting with another calf using the head or any other part of the body, like hitting or pushing."

Defecation/Urination – Suggestion: "Elimination of faces or urine from the body" (see Zambelis et al., 2019, https://doi.org/10.3168/jds.2018-15766).

L128. Remove “…was also registered…” and insert a “,” before “the”.

L129. Remove “;”               

L130. Remove “…presence/absence of...”.

L146-147. The methodologies used for data analyses are not clear. Please explain with more detail how the data analyses were carried out.

L442-445. Redundant. This is expected, isn't it? Culled cows are usually culled because they are facing a health/reproductive problem. Clarify

Reviewer 2 Report

General comments: This study reports a description of the effects of  transport stress in calves marketed through Chilean auction.

The paper is clear and well written. Overall, this is an interesting manuscript that provides additional information on the effects of transport stress, indicating the related effects of good or prohibited practices on  the behavioral indicators, according to Chilean law.

With the objective of this study, this article gave additional new information on the effect of transport stress on the behavioral pattern.

Some important improvements must be made to the article:

The work interesting and valuable - also for practical use, however some changes/improvement/ additional information needed.

The title is redundant and will be changed, whit an additional behavior evaluation

Some comments added in the manuscript, some references missing (few listed below).
I recommend adding some data about environmental conditions in the study area, some results for compared seasons.

The paper contains useful data which deserve to be published. However, the whole “transport conditions” is too complex in relation to the findings and should concentrate on the significant experimental results achieved. The results should be discussed in a focused manner towards the conclusions.

The comparative data in the control with respect the transport groups could identify the ameliorative physiological reactions, with possible animal welfare and well being implications.

What is the hypothesis of research?

What is the cost-and-benefit analysis in the calves’ health, well-being and welfare?

Some minor references missing, e.g. (you can find more when rewriting your paper)

Fazio E., Medica P., Cravana C., Cavaleri S., Ferlazzo A. (2012)

Effect of temperament and prolonged transportation on endocrine and functional variables in young beef bulls.

VETERINARY RECORD, 171:644.

Effect of long distance road transport on thyroid and adrenal function and haematocrit values in

Limousin cattle: influence of body weight decrease”

Fazio E, Medica P, Alberghina D, Cavaleri S, Ferlazzo A

 Vet. Res. Commun., 2005, 29, 713-719.

In my opinion, the paper while important in subject, is acceptable for publication after minor revisions.
